# Attenuating RNA Viruses with Expanded Genetic Codes to Evoke Adjustable Immune Response in PylRS-${\mathrm{tRNA}}_{\mathrm{CUA}}^{\mathrm{Pyl}}$ Transgenic Mice

**DOI:** 10.3390/vaccines11101606

**Published:** 2023-10-17

**Authors:** Zhetao Zheng, Xuesheng Wu, Yu Wang, Xu Yang, Hongmin Chen, Yuxuan Shen, Yuelin Yang, Qing Xia

**Affiliations:** State Key Laboratory of Natural and Biomimetic Drugs, Department of Chemical Biology, School of Pharmaceutical Sciences, Peking University, Beijing 100191, China; zhetaozheng@pku.edu.cn (Z.Z.); xuesheng96@pku.edu.cn (X.W.); yuwang9477@pku.org.cn (Y.W.); yangxu@bjmu.edu.cn (X.Y.); hongminchen@pku.edu.cn (H.C.); shenyuxuan2021@pku.edu.cn (Y.S.); yangyuelin@bjmu.edu.cn (Y.Y.)

**Keywords:** genetic code expansion, virus engineering, defective interfering particles, replication-incompetent virus, virus evolution

## Abstract

Ribonucleic acid (RNA) viruses pose heavy burdens on public-health systems. Synthetic biology holds great potential for artificially controlling their replication, a strategy that could be used to attenuate infectious viruses but is still in the exploratory stage. Herein, we used the genetic-code expansion technique to convert Enterovirus 71 (EV71), a prototypical RNA virus, into a controllable EV71 strain carrying the unnatural amino acid (UAA) Nε-2-azidoethyloxycarbonyl-L-lysine (NAEK), which we termed an EV71-NAEK virus. After NAEK supplementation, EV71-NAEK could recapitulate an authentic NAEK time- and dose-dependent infection in vitro, which could serve as a novel method to manipulate virulent viruses in conventional laboratories. We further validated the prophylactic effect of EV71-NAEK in two mouse models. In susceptible parent mice, vaccination with EV71-NAEK elicited a strong immune response and protected their neonatal offspring from lethal challenges similar to that of commercial vaccines. Meanwhile, in transgenic mice harboring a PylRS-${\mathrm{tRNA}}_{\mathrm{CUA}}^{\mathrm{Pyl}}$ pair, substantial elements of genetic-code expansion technology, EV71-NAEK evoked an adjustable neutralizing-antibody response in a strictly external NAEK dose-dependent manner. These findings suggested that EV71-NAEK could be the basis of a feasible immunization program for populations with different levels of immunity. Moreover, we expanded the strategy to generate controllable coxsackieviruses for conceptual verification. In combination, these results could underlie a competent strategy for attenuating viruses and priming the immune system via artificial control, which might be a promising direction for the development of amenable vaccine candidates and be broadly applied to other RNA viruses.

## 1. Introduction

RNA viruses account for the majority of human viral infections [1] and remain difficult to control despite the availability of several working vaccine strategies [2,3,4]. While synthetic biology holds great potential for biomedical engineering, it is limited in accessing vector design for gene therapy [5,6,7,8], which renders it far from practical in vaccine production [9,10]. By integrating advances in reverse genetics [11,12] with insights from system vaccinology [13,14], viral-engineering methods may be adapted to effective and smart vaccine design in a novel way.

Replication-incompetent viruses that harbor artificial amber codons in their genome and propagate using genetic-code expansion technology [15,16] elicit robust immunity in animal models and improve upon traditional vaccine approaches [17,18]. Nevertheless, the decisive step is the incorporation of an unnatural amino acid (UAA) into the nascent viral polypeptide at desired positions, which are selected by massive random mutations with undefined rules [18], impacting the amenability of this approach to emerging viruses. Moreover, UAA modification is a switch that enables artificial control of virus replication [19,20], hinting at a potential strategy of precisely controlling viral replication in vivo to imprint desired antibody landscapes.

We applied the genetic-code expansion technique to modify *Enterovirus* 71 (EV71) as a model to (a) collect data that would define rules for UAA incorporation and (b) test whether the optimized EV71-UAA would require the conservation of the full infectious form for immunogenicity and replication under artificial control in vitro and in vivo for safety [21,22]. We further exploited the virus to evoke strong and adjustable antibody responses in normal and transgenic mice harboring PylRS-${\mathrm{tRNA}}_{\mathrm{CUA}}^{\mathrm{Pyl}}$ pairs as a potential approach to vaccinating susceptible individuals with differing levels of immunity [23].

## 2. Results

### 2.1. In Vitro UAA-Controllable RNA Virus-Packaging and -Production System

In this study, we chose EV71, the major pathogen in hand-foot-mouth disease (HFMD) in children and infants [24,25], as a model virus to test the feasibility of a production pipeline for viruses with expanded genetic codes. An amber codon artificially introduced into the viral genome renders the virus replication incompetent in conventional cells but recovers replicative potential using UAA incorporation machinery, which is composed of three biorthogonal elements: aminoacyl-tRNA synthetases (aaRS), transfer RNA (tRNA), and UAA (Figure 1A). To improve the compatibility of the incorporation machinery with the production of RNA viruses carrying UAA, we designed a two-round viral-production pipeline: (a) viral RNA into which an amber codon was introduced was transfected into human embryonic kidney 293T (HEK293T) cells integrated with well-established orthogonal MmPylRS/${\mathrm{tRNAMm}}_{\mathrm{CUA}}^{\mathrm{Pyl}}$ pairs (NAEK-HEK293T cells) for first-round viral packaging (Figure 1A, left); (b) propagation of a NAEK-Vero stable cell line for high-yield viral production in the presence of NAEK (Figure 1A, right). Given that the NAEK-HEK293T stable cell line was previously reported [18], we first sought to establish the NAEK-Vero line by applying the *Desulfobacula toluolica* (Tol2) transposon to integrate the orthogonal pair into the Vero-E6 cells genome (Appendix A). We confirmed the successful expression of MmPylRS, tRNA^Pyl^, and NAEK-dependent GFP^39TAG^ carrying amber-codon replacements in 39 sites, facilitating the selection of cell lines with efficient UAA translation machinery (Figure 1B,C). In addition, we observed that the machinery’s introduction did not significantly affect cell proliferation in either line, which was beneficial for viral packaging and propagation (Figure 1D).

After establishing appropriate viral-packaging systems in the stable cell lines, we evaluated the effect of the NAEK system’s incorporation on gene expression via whole-transcriptome analysis (WTA). Whole-transcriptome plots and principal-component analysis (PCA) revealed that the cell line with engineered systems had gene expression patterns similar to those of HEK293T cells, which differed significantly from those of EV71-infected HEK293T cells (Figure 1E,F). The Venn diagram (Figure 1G) shows that the number of differentially expressed genes (DEGs) in EV71-infected HEK293T cells was significantly higher than that in the stable cell line ± NAEK groups. Hierarchical-clustering analysis of DEGs showed two main clusters, which are displayed in the heatmap, further indicating that the stable cell line ± NAEK groups had DEGs similar to those of HEK293T cells, which in turn were very different from those of EV71-infected HEK293T cells (Figure 1H). Gene Ontology (GO) terms and Kyoto Encyclopedia of Genes and Genomes (KEGG) signaling pathways further indicated their common application of stable cell line ± NAEK groups for generating artificial viruses from other extant RNA viruses (Figure 1I and Appendix A). The WTA results were similar in the NAEK-Vero stable cell line (Appendix A). The UAA system displayed the same DEG expression patterns, GO terms, and KEGG signaling pathways as in HEK293T cells, indicating that the effect of UAA incorporation was far less than that of viral infection and that the system was appropriate for controllable viral production.

### 2.2. Parallel Screening Revealed Rules for NAEK Incorporation

A guideline for identifying optimal amber-codon introduction into the viral genome based on reliable NAEK incorporation machinery is pivotal to an effective NAEK-controllable viral replication system. Therefore, we profiled the entire viral genome by generating different EV71-NAEK viruses containing mutant amber codons and observed the varying viral titers (Figure 2A). Although all encoding genes except 2B contributed to at least one mutant codon (2 of 10 codons in VP4, 1 of 4 in VP2, 1 of 4 in VP3, 5 of 10 in VP1, 1 of 5 in 2A, 0 of 3 in 2B, 3 of 10 in 2C, 1 of 4 in 3A, 1 of 3 in 3B, 5 of 5 in 3C, and 9 of 10 in 3D) whose NAEK-controllable virus could be successfully packaged and propagated, we noted that viruses harboring amber codons in the viral 3C or 3D genes showed prevalently higher viral titers (Figure 2A and Appendix A). To assess the features of incorporated positions that could affect viral titer, we further tested 15 and 20 mutant codons located in particular EV71-3C and 3D genes, respectively (Figure 2B,C). We thus generated 9 and 11 variants, respectively, with virions of 10^4^–10^6.5^ TCID_50_/mL, and clearly observed NAEK-dependent cytopathic-effect (CPE) phenotypes as expected (Figure 2D and Appendix A).

To confirm the genetically stable production of EV71-NAEK, we performed additional selection by passaging all 20 strains of EV71-3C/3D-NAEK virus in stable NAEK-Vero cells for 20 rounds. Unexpectedly, the loss of NAEK-dependent CPE phenotype was observed in 3D-N365NAEK strains, consistent with the high escape frequencies over 20 generations. Meanwhile, 3D-E105NAEK viruses were stably inherited in respective strains, with relatively low escape frequencies (3.6 × 10^−7^ to 5.1 × 10^−6^) over passaging verified by next-generation sequencing (Figure 2E,F), indicating the varied genetic stability at different loci in the viral genome as previously reported [26,27]. To optimize the guideline for position identification, we annotated the escape frequency with conservation analysis of the corresponding amino acid residues of viral proteins by ConSurf calculations and found that a low conservation of amino acid was also correlated with high escape frequency at individual mutant codons (Figure 2F), suggesting a delicate balance between viral titer and escape frequency.

These results indicated that incorporating UAA to replace amino acids with similar properties (NAEK to lysine or glycine) and appropriate conservation could increase the production efficiency and genomic fidelity of controllable viruses (Appendix A). Of the 35 incorporation positions screened in this study, 30 residues were located on the surface of 3C/3D, and the other five were inaccessible to the solvent (Appendix A). Furthermore, surface sites might be more open to NAEK with respect to amino acid similarity (Appendix A). Other factors, such as residue-residue interaction or steric effects, also affected overall package efficiency to some extent. Therefore, we performed a logistic-regression analysis (Appendix A), the results of which indicated that viral packaging correlated positively and significantly (W4 > 10, *p* < 0.05) with amino acid similarity. Therefore, when the original amino acid at a certain position resembled NAEK, the respective mutant amber codon tended to be better read-through by the corresponding system, rendering controllable viral production more possible. In addition, the NAEK system was also sensitive to the conservation level (W3 = −3.6, *p* < 0.05), indicating that NAEK could be better incorporated into the less conserved site of a viral protein. 

### 2.3. Prediction and Validation of UAA-Controllable Virus Design

We tested whether the aforementioned logistic function could be applied to other RNA viruses to guide mutant amber-codon selection. The main types of *Enterovirus* that account for human infections, including the enteroviruses EV71 and EVD68 and the coxsackieviruses CA6, CA10, and CA16, comprised the P1, P2, and P3 genomes, respectively encoding the 3AB, 3C, and 3D proteins (Appendix A). We thereby managed to expand our strategy to the abovementioned RNA viruses, demonstrating the feasibility of logistic function. We found similar NAEK-dependent CPE formation and viral RNA copies of those viruses compared with wild-type (WT) viruses (Appendix A).

For EV71, we selected EV71-3D-E105NAEK, with one codon encoding Glycine in the 3D^P^°^l^ protein, for amber-codon replacement (3D-E105^TAG^), and attempted to survey its viral characteristics. After EV71-3D-E105NAEK infection of both cell lines, we observed the CPE phenotype, indicating that successful production and widespread propagation of EV71-3D-E105NAEK was NAEK dependent (Figure 2G). Compared with that of WT EV71, the growth curve of EV71-3D-E105NAEK further demonstrated that NAEK-controllable viral replication was well established in vitro (Figure 2H). Moreover, transmission electron microscopy (TEM) showed no significant differences in morphologies, implying that EV71-3D-E105NAEK maintained the original phenotype of parental EV71 (Figure 2I). Owing to its combined high viral titer and low escape frequency, we chose EV71-3D-E105NAEK as a model strain for further study in vivo.

### 2.4. Efficacy, Safety, and Protective Evaluation of EV71-3D-E105NAEK as a Vaccine Candidate In Vivo

Next, we evsaluated the in vivo safety and immunogenicity of intraperitoneally (i.p.) injected EV71-3D-E105NAEK in adult BALB/c mice, using commercial Sinovac-EV71 vaccine as a positive control (Figure 3A). No adult mice died from EV71-3D-E105NAEK or showed obvious body weight (BW) loss or other health issues due to this injection (Figure 3B,C). We detected no significant differences in viral load in the brain or skeletal muscle between the EV71-3D-E105NAEK and Sinovac-EV71 groups, and even less viral load in the small intestine was observed in EV71-3D-E105NAEK mice than in Sinovac-EV71 mice, indicating the good in vivo safety of EV71-3D-E105NAEK as a vaccine candidate (Figure 3D). Two weeks after single- or double-dose immunization, both the EV71-3D-E105NAEK and Sinovac-EV71 vaccines induced robust serum immunoglobulin M (IgM) and antiviral protein 1 (VP-1) IgG (Figure 3E,F). To determine the potential protective efficacy of EV71-3D-E105NAEK in neonatal mice, we injected a lethal EV71 strain (SD059) i.p. into the 4-day-old offspring of immunized BALB/c mice. All of these neonatal mice—in contrast with those in the vehicle group that succumbed within 14 days—survived after immunization with EV71-3D-E105NAEK and Sinovac-EV71 vaccines (Figure 3G), with no significant BW loss (Figure 3H). In addition, viral load in the small intestine, brain, and spinal cord in both the EV71-3D-E105NAEK and Sinovac-EV71 groups was significantly lower than in the vehicle group (Figure 3I), indicating the protective efficacy of EV71-3D-E105NAEK in neonatal mice. Additionally, immunostaining and hematoxylin and eosin (H&E) staining of the small intestine showed no EV71 viral replication or pathological changes in the neonatal offspring of immunized BALB/c mice compared with the vehicle group (Figure 3J,K). In brief, our method could produce a reliable EV71-3D-E105NAEK vaccine candidate, comparable to the commercial vaccine, in a mouse model.

### 2.5. In Vivo UAA-Controllable RNA Virus Evoked Adjustable Immune Response in PylRS-${tRNA}_{CUA}^{Pyl}$–Transgenic Mice

To investigate NAEK-controllable RNA viral replication in vivo with safeguards, we constructed transgenic mice stably harboring MmPylRS/${\mathrm{tRNAMm}}_{\mathrm{CUA}}^{\mathrm{Pyl}}$ via genome editing as previously described [19,28,29,30] to complement the in vitro UAA-controllable system (Figure 4a). The expression of MmPylRS, ${\mathrm{tRNAMm}}_{\mathrm{CUA}}^{\mathrm{Pyl}}$, and GFP^39TAG^ was confirmed in the small intestines of transgenic mice (Figure 4B,C and Appendix A). Next, we inoculated the transgenic mice with EV71-3D-E105NAEK and administered daily i.p. injections to these mice of NAEK at different doses (−, no, 0 mg; +, low, 16 mg; ++, medium, 25 mg; +++, high, 50 mg) according to estimated bioavailability in the mice’s serum and small intestines (Appendix A). Three days later, we detected increased viral-RNA levels in an NAEK dose-dependent manner in the small intestine, brain, and spinal cord (Figure 4D). Meanwhile, accumulating EV71-3D^P^°^l^ proteins were clearly observable by immunostaining and the pathological changes to the intestine were shown by H&E staining (Figure 4E,F). No adult mice died from any NAEK or EV71-3D-E105NAEK dose; no obvious BW loss or other health issue in the low- and medium-dose groups was observed, as opposed to the high-dose group (Figure 4G,H). Serum IgM and anti-VP1 IgG were robustly induced in a dose-dependent manner, even in the low-dose group (Figure 4I), which was consistent with the boost dose of commercial Sinovac-EV71 vaccine (Figure 3E,F), indicating the optimal dose for enhanced protective efficacy and few health concerns. These results demonstrated that the replication of EV71-3D-E105NAEK was NAEK controllable and dose-dependent in transgenic mice, further implying that EV71-3D-E105NAEK has great potential as a safe vaccine candidate. A single dose of EV71-3D-E105NAEK followed by optional UAA uptake could be the basis of a feasible immunization program to provoke optimal immune response in susceptible individuals with distinct levels of immunity, instead of injection of boost and even booster doses.

## 3. Discussion and Conclusions

The success of this viral-engineering strategy defined guidelines for UAA incorporation and viral generation that could be broadly expanded to other RNA viruses in this study provided a potential solution to the immediate quandary of vaccination by harnessing EV71-NAEK to elicit strong immune responses that were adjustable via artificial NAEK administration. For UAA-mediated viral engineering, although genetic codon expansion technology has been employed to modify virus-like particles [31,32], engineer viral vectors for gene therapy [33,34], and probe the biology of viruses [35,36], UAA-incorporated sites are generally selected by random mutations within targeted viral proteins, a traditional screening method for protein engineering [37], and the lack of defined rules considering protein modification, virus assembly, and reversion potential of amber codon. Moreover, engineered viruses developed in previous studies were always packaged in HEK293T cells with transiently or stably expressed PylRS/tRNA pairs [18], which is appropriate for the generation of common viral vectors (e.g., adeno-associated viruses [AAVs], lentiviruses) but not for most viruses’ propagation, especially in vaccine development [38].

Herein we reported that the established two-round viral-engineering system, consisting of NAEK-HEK293T cells for transfection and NAEK-Vero cells for propagation, could generate UAA-engineered viruses for main intestinal pathogens, including enteroviruses and coxsackieviruses, with high efficiency. Based on parallel screening in the EV-71 genome, we took chemical similarity, gene conservation, viral-protein property, and engineered viral titer into consideration when defining rules for UAA incorporation via logistic regression analysis.

The final principle can be adapted to the diverse set of viable UAAs [39] to generate engineered viruses with different modes of incorporation. Beyond the control of their replication, viral phenotypes can be simultaneously modified by the versatile side chain of UAA to jointly produce several desirable properties, such as improved cell tropisms and manufacturability [40,41]. This is substantially more challenging than traditional methods of genetic engineering.

Additionally, UAA-engineered viruses restore the full infectious form for direct priming of the immune system, and they contain the full genome, which can recapitulate RNA replication and protein translation without virion assembly in vivo, like RNA-based vaccines in principle [42]. Therefore, we chose EV71-NAEK to validate that its safety and efficacy were comparable with those of currently available vaccines. In our mouse model, few viral RNA copies were detected even in theEV71–NAEK–vaccinated group. However, as maturing biotechnologies and extensive immunological discoveries advance the development of next-generation vaccines, the issue remains that policy-consistent vaccinations would be improper for individuals with poor immunity, especially children [43], older adults [44], and pregnant women [45]. The alternative solution is to choose a vaccine that might induce weak immune responses or change the dose and interval of vaccination; both strategies would fall short of the desired protective efficacy and would be impractical in large-scale applications.

Our study highlighted that the amendable EV71-NAEK could infect and replicate in vivo to elicit immune responses under control. To our knowledge, this is the first demonstration to date of vaccination with a controllable pathogen to evoke adjustable antibody landscapes. Compared with those of commercial vaccines, the absolute antibodies varied from 0.5-fold to approximately 2.5-fold in an external NAEK dose-dependent manner, suggesting that efficient protection can be achieved via an optimal dose of UAA according to the patient’s level of immunity. Currently, UAA-engineered virus retaining the full infectious form presents diverse antigens that, within artificial thresholds, would efficiently prime the human immune system with transient delivery of PylRS-${\mathrm{tRNA}}_{\mathrm{CUA}}^{\mathrm{Pyl}}$ pairs and oral UAA uptake [19], a novel and convenient vaccine strategy for combating emerging viruses. More generally, with advances in synthetic biology in smart materials and cell therapies [46], the development of synthetic gene circuits for sensing immune responses and outputting UAAs to control viral replication could become the basis of a potential immunization program.

In conclusion, we systematically explored a novel strategy to convert infectious viruses with unnatural amino acids to achieve controllable viral replication and elicit an adjustable immune response in vivo, potentially creating a next-generation vaccine candidate. This study has laid the foundation for the bioengineering design of smart vaccine candidates responsive to immunity, drawing on both synthetic biology and viral engineering.

## Figures and Tables

**Figure 1 vaccines-11-01606-f001:**
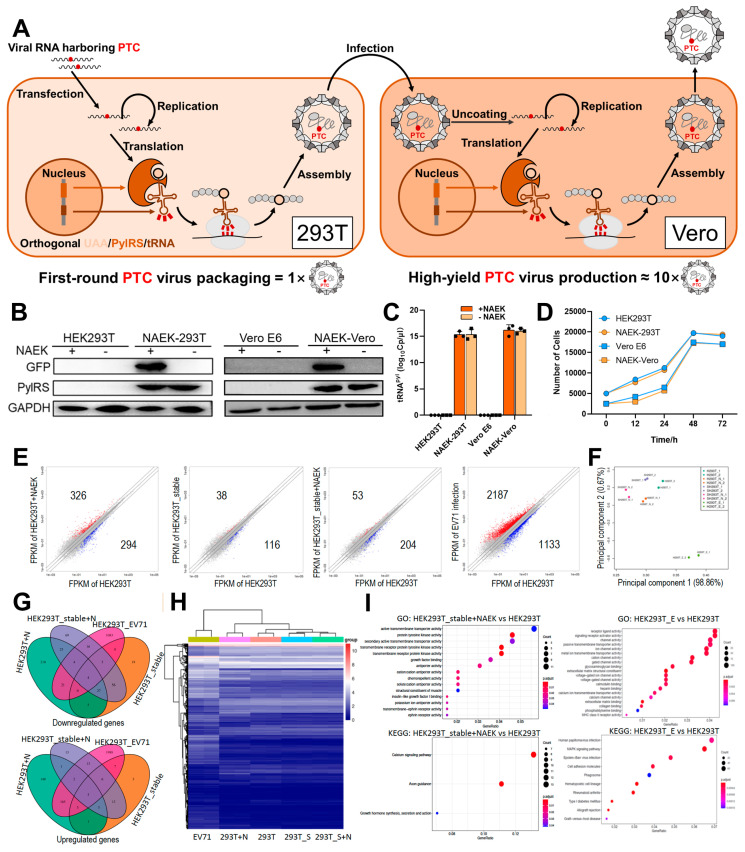
Construction and identification of UAA packaging system to generate EV71-NAEK virus. (**A**) Schematic illustration of EV71-NAEK viral packaging and production. An artificial amber codon was introduced into the viral genome to create a replication-incompetent virus using engineered HEK293T cells harboring a UAA incorporation system composed of orthogonal PylRS, tRNA^Pyl^, and the relevant UAA (NAEK in this study). The UAG was read through by the UAA incorporation system, generating the EV71-NAEK virus, which was then produced on a large scale by Vero-E6 cells harboring the UAA system. (**B**) The characterization of the UAA packaging system by WB analysis. (**C**) Reverse transcription polymerase chain reaction (RT-PCR) validation of the tRNA^Pyl^ transcripts. Data are represented as mean ± standard deviation (SD; n = 3). (**D**) The effect of engineered HEK293T and Vero-E6 cells harboring the UAA incorporation system on cell proliferation. (**E**) WTA of HEK293T cells, the engineered packaging systems (HEK293T_stable), and EV71-infected HEK293T cells. The plots show whole-transcriptome fragments per kilobase of exon per million fragments mapped (FPKM). Red dots indicate upregulated genes; blue dots are downregulated genes. Two biological replicates were used per sample. (**F**) PCA of HEK293T±NAEK, HEK293T_stable (SHEK293T) ±NAEK and EV71-infected HEK293T cells (HEK293T_E). (**G**) Venn diagram showing significant overlap (*p* < 0.005) of DEGs among HEK293T cells, engineered packaging systems, and EV71-infected HEK293T cells. (**H**) Hierarchical clustering and heatmap analysis of DEGs in HEK293T cells, the engineered packaging systems, and EV71-infected HEK293T cells. (**I**) Representative GO terms and KEGG enrichment analysis of the engineered HEK293T+NAEK and EV71-infected HEK293T cells compared with HEK293T cells.

**Figure 2 vaccines-11-01606-f002:**
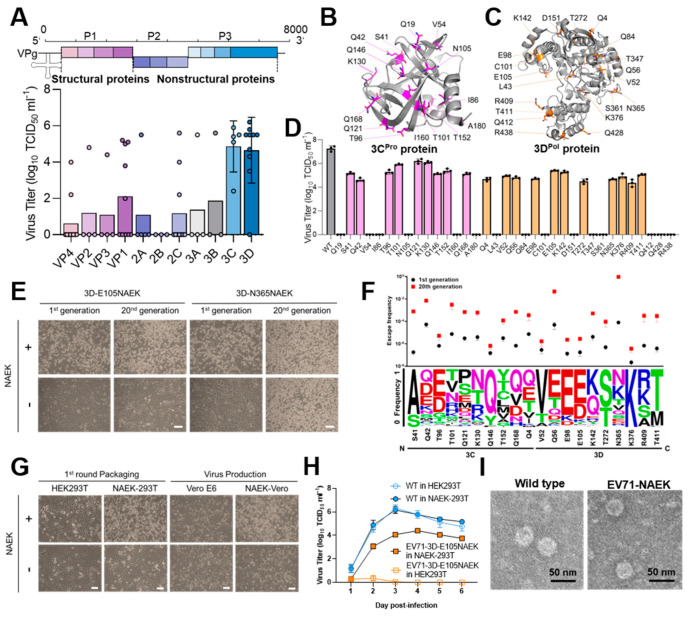
Screening for optimal incorporation sites. (**A**) Optimal amber insertion genes (including VP1–VP4, 2A–2C, and 3A–3D proteins) were discovered by profiling the viral genome, which turned out to be the gene for 3C/3D protein. (**B**–**D**) We tested 35 locations in the gene of variable, average, and conserved domains in EV71-3C/3D protein and identified the optimal mutant site for the UAA system to read through. That site was 3D-E105NAEK, due to its relevant high packaging efficiency and viral titer. (**E**) Verification of EV71-3D-E105NAEK’s genetic stability, reflected by UAA-dependent CPE formation after 1 and 20 passages in transgenic HEK293T-tRNA/PylRS/GFP^39TAG^ cells. EV71-3D-N365NAEK was used as a control due to reverse mutation in passage 20. Scale bars, 50 μm. (**F**) Relationship between conservation of amino acid and escape frequency after 1 and 20 passages. Among screened sites, 3D-E105 was relatively conserved. (**G**) CPE of EV71-3D-E105NAEK packaged with the HEK293T-NAEK system and produced by the VeroE6-NAEK system in the presence or absence of NAEK. Scale bars, 50 μm. (**H**) Growth kinetic curves of EV71-3D-E105NAEK generated by the NAEK system in the presence or absence of NAEK at indicated time points. Data are represented as mean ± SD (n = 3). (**I**) Morphologies of the EV71-3D-E105NAEK and WT EV71 viruses as shown by TEM. Scale bars, 50 nm.

**Figure 3 vaccines-11-01606-f003:**
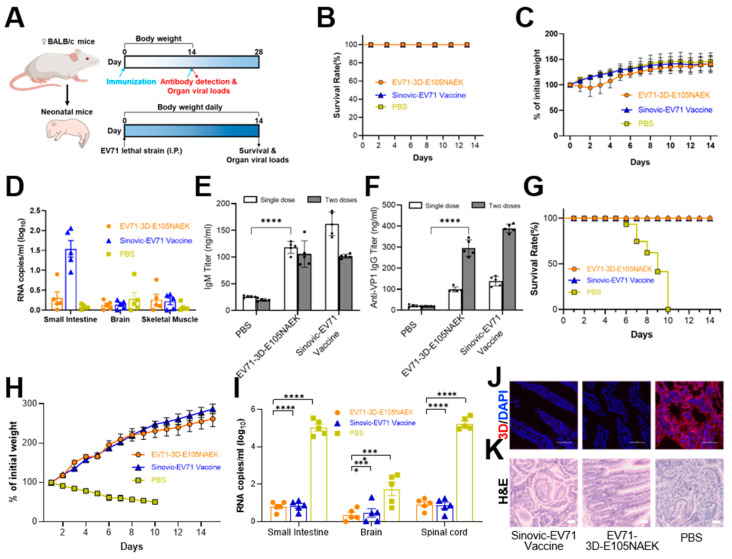
Evaluation of the EV71-3D-E105NAEK virus’s safety, immunogenicity, and protection levels. (**A**) Schematic of study design. To evaluate safety and immunogenicity, 4-week-old female BALB/c mice were first injected i.p. with EV71-3D-E105NAEK for 14 days. To evaluate protection, the 4-day-old offspring of immunized BALB/c mice received i.p. injections of a lethal EV71 strain (SD059) for 14 days. (**B**,**C**) Survival rates and BW changes in BALB/c mice. (**D**) Viral RNA copies are detected in the small intestine, brain, and skeletal muscle. (**E**,**F**) IgG and IgM titers were detected before and after one or two doses of immunization. (**G**,**H**) Survival rates and BW changes in neonatal child BALB/c mice. (**I**) Detection of viral RNA copies in neonatal-mouse tissues 3 days after inoculation with 10^5^ PFU of the EV71 virus, EV71-3D-E105NAEK, or Sinovac-EV71 vaccine. (**J**,**K**) Fluorescent images and H&E staining of small intestines of neonatal mice at day 14. Scale bars, 200 μm. Data in (**B**–**I**) are presented as mean ± SD (n = 5). Two-way analysis of variance (ANOVA) and Tukey’s test were performed. * *p* < 0.05, *** *p* < 0.001, **** *p* < 0.0001.

**Figure 4 vaccines-11-01606-f004:**
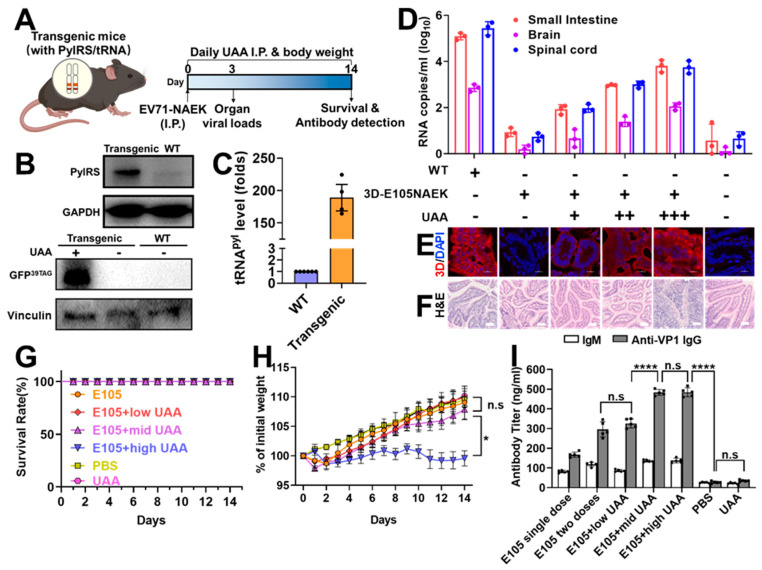
Immune response to UAA-controllable EV71-3D-E105NAEK in transgenic mice. (**A**) Transgenic mice harboring the UAA incorporation system were first injected i.p. with EV71-3D-E105NAEK for 14 days in the presence or absence of UAA. (**B**) WB analysis of PylRS and GFP^39TAG^ expression in transgenic mice. (**C**) Quantitative RT-PCR (qRT-PCR) analysis of tRNA^Pyl^ levels in transgenic mice. (**D**) Detection of viral-RNA copies in the small intestine, brain, and spinal cord after inoculation with 10^5^ PFU EV71-3D-E105NAEK in the presence or absence of UAA. (**E**,**F**) Fluorescent images and H&E staining of the small intestines of neonatal mice on day 14. Scale bars, 200 μm. (**G**,**H**) Survival rates and BW changes in transgenic mice after inoculation with EV71-3D-E105NAEK. (**I**) IgG and IgM titers were detected after immunization with EV71-3D-E105NAEK in the presence of low-, medium-, or high-dose UAA, indicating the UAA dose-dependent manner. Data (**D**,**G**–**I**) are presented as mean ± SD (n = 5). Two-way ANOVA and Tukey’s test were performed. * *p* < 0.05, **** *p* < 0.0001. Not significant (n.s.) stands for *p* ≥ 0.05.

## Data Availability

The data presented in this study are available in the article and Appendix A.

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
