# Peer review of "Attenuating RNA Viruses with Expanded Genetic Codes to Evoke Adjustable Immune Response in PylRS- Transgenic Mice"

_vaccines, 2023, doi:10.3390/vaccines11101606_

Round 1
Reviewer 1 Report
In this study, the authors test a novel potential vaccination strategy based on the use of the UAA codon to insert an unnatural amino acid in defined sites in the Enterovirus 71 (EV71) genome. They showed that this virus (EV71-NAEK) could replicate in an authentic manner making it amenable to manipulation to control the immune response in different systems. They then go on to show that this virus can elicit a robust antibody response and prevent weight loss and death in immunized animals. Subsequently, they attempt to test the strategy with other RNA viruses, including coxsackieviruses and SARS-CoV-2.
This is a well designed and effectively carried out study. The data make a strong case that the attenuation of RNA viruses through the insertion of unnatural amino acids is an approach that carries a great deal of potential for vaccine design.
The attempt to extend the findings with EV71 to other RNA viruses, SARS-CoV-2 and coxsackievirus, is commendable. However, especially for SARS-CoV-2, partially due to the BSL-3 properties of the virus, the feasibility of the approach is tested only minimally, i. e. by inserting the amber codon into its nucleocapsid (N) protein genes. While I appreciate that the authors wanted to extend their findings to more pathogenic viruses, the data shown represent only a rather cursory attempt to test the hypothesis. In my opinion, the brief attempt at extending the approach to these other two viruses should be deleted from the manuscript, as it is not extensive enough to really confirm anything.
English language is fine.
Author Response
We thank the editor and reviewers for the positive comments, which are very valuable to improve the paper. Our point-by-point responses to every comment are presented in the revised manuscript and all revised texts or figures are highlighted with yellow color in the manuscript. The attached PDF represents major changes according to the reviewers’ comments.

Reviewer 2 Report
The authors used a genetic code expansion technique to convert Enterovirus 71 (EV71) into a controllable EV71 strain carrying the unnatural amino acid (UAA) Nε-2-azidoethyloxycarbonyl-L-lysine 15 (NAEK). Vaccination with EV71-19 NAEK elicited a strong immune response. EV71-NAEK 22 evoked an adjustable neutralizing-antibody response in transgenic mice harboring a 21 PylRS-tRNAPyl CUA pair. This immunization approach has potential, and the experiments are well-done.
Lines 166-168 Please explain further about the high escape frequency of 3D-N365NAEK strains and link it to the description of figure 2E and F. Maybe a few more sentences of explanation would be helpful. I'm not sure that the description will be adequate for most readers.
Author Response

(The authors gave the same response as above.)

Reviewer 3 Report
It is an interesting study showing the effect of engineering RNA viruses on the induction of immune responses in mice. The authors developed new approach that control the infectivity of virus by introducing unnatural amino acids to achieve controllable viral replication and elicit an adjustable immune response.
The manuscript is well designed and had an acceptable flow.
Few major points
1) Ethical approval for animal study should be obtained in this study and mentioned in the manuscript.
2) Original uncropped gel for WB are required to be uploaded.
Language is ok
Author Response

(The authors gave the same response as above.)

Round 2
Reviewer 3 Report
No further comments
Moderate language editing